# Mitochondrial Respiration Changes in R6/2 Huntington’s Disease Model Mice during Aging in a Brain Region Specific Manner

**DOI:** 10.3390/ijms21155412

**Published:** 2020-07-30

**Authors:** Johannes Burtscher, Alba Di Pardo, Vittorio Maglione, Christoph Schwarzer, Ferdinando Squitieri

**Affiliations:** 1Department of Pharmacology, Medical University of Innsbruck, 6020 Innsbruck, Austria; 2IRCCS, Neuromed, 86077 Pozzilli, Italy; dipardoa@hotmail.com; 3Huntington and Rare Diseases Unit, Fondazione IRCCS Casa Sollievo della Sofferenza Research Hospital, 71013 San Giovanni Rotondo, Italy; f.squitieri@css-mendel.it

**Keywords:** Huntington’s disease, aging, neurodegeneration, mitochondria, oxidative phosphorylation, respiration

## Abstract

Mitochondrial dysfunction is crucially involved in aging and neurodegenerative diseases, such as Huntington’s Disease (HD). How mitochondria become compromised in HD is poorly understood but instrumental for the development of treatments to prevent or reverse resulting deficits. In this paper, we investigate whether oxidative phosphorylation (OXPHOS) differs across brain regions in juvenile as compared to adult mice and whether such developmental changes might be compromised in the R6/2 mouse model of HD. We study OXPHOS in the striatum, hippocampus, and motor cortex by high resolution respirometry in female wild-type and R6/2 mice of ages corresponding to pre-symptomatic and symptomatic R6/2 mice. We observe a developmental shift in OXPHOS-control parameters that was similar in R6/2 mice, except for cortical succinate-driven respiration. While the LEAK state relative to maximal respiratory capacity was reduced in adult mice in all analyzed brain regions, succinate-driven respiration was reduced only in the striatum and cortex, and NADH-driven respiration was higher as compared to juvenile mice only in the striatum. We demonstrate age-related changes in respirational capacities of different brain regions with subtle deviations in R6/2 mice. Uncovering in situ oxygen conditions and potential substrate limitations during aging and HD disease progression are interesting avenues for future research to understand brain-regional vulnerability in HD.

## 1. Introduction

Some of the most prevalent cellular aging paradigms have mitochondrial dysfunction at their core, such as the free radical theory of aging [1,2] or the related mitochondria [3,4,5] or mitohormesis paradigms [6] of aging. In very simplified terms, they posit that the wear and tear of mitochondrial activity, involving the production of reactive oxygen species, which in turn modify mitochondrial components, is integral for the aging processes. In line with these theoretical frameworks, moderate reductions of mitochondrial respiration have the potential to increase the life-span of various species, such as *C. elegans* [7,8,9], *D. melanogaster* [10], and mice [11,12], supporting the theory of a correlation of the metabolic rate and life span [13]. On the other hand, age-related alterations in mitochondrial functions are strongly associated with neurodegenerative diseases [14]. Strikingly, in most neurodegenerative diseases, specific brain regions are selectively affected [15]. Still puzzling for research, different brain regions succumb to neurodegeneration in different neurodegenerative diseases. Brain atrophy is most striking in the caudate-putamen (in rodent brain also called striatum) in Huntington’s Disease (HD), with striatal medium spiny neurons being the most vulnerable cell type [16], although also other brain regions are affected as well. Like in other neurodegenerative diseases, mitochondrial deficits are a core pathology in the HD affected brain. This became apparent with the observation of structurally compromised mitochondria [17], impaired mitochondrial respiration in the caudate nucleus [18] and defects in multiple components of the electron transfer system [19] in post-mortem Huntington’s Disease (HD) patient brains, as well as from the reproduction of HD-resembling selective striatal neurodegeneration with the mitochondrial Complex II inhibitor, 3-nitropropionic acid (3-NP) [20]. The disease-causing mutated Huntingtin (Htt) protein is furthermore known to interact with mitochondria and to impair Complex II [21].

R6/2 mice, which overexpress human N-terminal Htt (animals of our study with around 160 glutamine repeats) are commonly used as a model for HD [22], but results on mitochondrial respiration in these mice are conflicting. Isolated mitochondrial Complex IV activity has been shown to be reduced in striatum and cortex of 12 weeks old R6/2 mice [23]. Studies of mitochondrial respiration of ex vivo brain tissues, however, suggest this to be of no significant functional consequence. While Aidt and colleagues [24] reported (small) respirational deficits (only in flux control ratios) using high resolution respiration in the striatum of 12 weeks old symptomatic R6/2 mice, no respirational impairments were observed by Hamilton and colleagues in 6–8-week-old symptomatic R6/2 mice [25].

Why specific brain regions in HD are particularly vulnerable, and how mitochondrial dysfunction may be involved in such vulnerabilities remains an enigma. We previously observed in brains of healthy, adult mice an intriguingly higher contribution of mitochondrial Complex II to total mitochondrial respiration in the striatum as compared to other brain regions [26] and proposed that the relative importance of Complex II specifically in the striatum together with Htt’s inhibitory effect on Complex II [21] may constitute the basis of the selective vulnerability of the striatum in HD.

On the basis of these findings, we hypothesize that mitochondrial respirational patterns might not only differ across brain regions in the adult brain but also depend on age. In addition, we hypothesize that during HD progression respirational patterns might change in a different fashion as compared to healthy aging.

To address these hypotheses, we study mitochondrial respiration in juvenile and adult wild-type mice and compare the results to respirational patterns in age-matched pre-symptomatic and symptomatic R6/2 mice.

## 2. Materials and Methods

### 2.1. Animals

Transgenic mice [strain name: B6CBA-tgN (HDexon1) 62Gpb/1J] with 160 (CAG) repeat expansions (R6/2 line) were obtained from Jackson Laboratories. Crossing with female B6CBA wild-type (WT) mice enabled the establishment of the animal colony. All experimental procedures were approved by the IRCCS Neuromed Animal Care Review Board and were conducted according to the 2010/63/EU Directive for animal experiments.

Female R6/2 mice and WT littermates were sacrificed at either 4 weeks of age (R6/2 pre-symptomatic) or at 10–11 weeks of age, when R6/2 mice display pronounced pathology (see main text).

### 2.2. High-Resolution Respirometry

Wild-type and R6/2 mice were sacrificed by cervical dislocation between 9 and 10 am in the morning. The striata, hippocampi, and motor cortex were quickly dissected on ice. After weighing the wet tissues, they were placed and washed in mitochondrial respiration medium MiR06Cr (4 °C). MiR06Cr contained 280 IU/mL catalase (Pesta and Gnaiger, 2012) and 20 mM creatine. The tissues were then homogenized in MiR06Cr (4 °C) using a pre-cooled glass potter (tight fit; WiseStir homogenizer HS-30E, witeg Labortechnik GmbH, Wertheim, Germany) at 1000 rpm, as described previously (Burtscher et al., 2015). Resulting homogenates of 1 mg/mL of ice-cold MiR06Cr were used for OXPHOS-analysis in duplicates.

Respiration was measured at 37 °C in the Oroboros O2k (Oroboros Instruments, Austria) (*n* = 4 per genotype and age category), as described previously [27]. Briefly, tissue homogenates were moved into calibrated Oxygraph-2 k (O2k, OROBOROS INSTRUMENTS, Innsbruck, Austria) 2 mL-chambers. Oxygen concentration (μM) as well as oxygen flux per tissue mass (pmol O_2_.s^−1^·mg^−1^) were assessed and recorded with DatLab software (OROBOROS INSTRUMENTS, Innsbruck, Austria). The oxygen concentration was set to 360–380 μM at the beginning of the experiment.

The non-phosphorylating, NADH (N)-linked LEAK-respiration (N*_L_*) was measured in the presence of the CI-linked substrates pyruvate (5 mM), malate (0.5 mM) and glutamate (10 mM). CI-linked respirational capacities (N*_P_*) were achieved by addition of 2.5 mM ADP. Then, 50 mM succinate (S) were added to induce CI and II substrate-linked respiration (NS*_P_*). Carbonyl cyanide m-chloro phenyl hydrazone (CCCP) mediated uncoupling yielded the maximum capacity of the electron transfer system (ETS, NS*_E_*). The inhibition of CI by rotenone (0.5 μM) allowed the assessment of succinate-driven ETS capacity (S*_E_*). Residual oxygen consumption (ROX) was measured after malonate and antimycin A addition and subtracted from mitochondrial respiratory states. The measurement of CIV activity was then performed on the same samples. After reoxygenation, ascorbate (2 mM) and *N*,*N*,*N*′,*N*′-Tetramethyl-p-phenylenediamine dihydrochloride (TMPD, 0.5 mM) were added. The chemical background was assessed by subsequent inhibition of CIV by sodium azide (100 mM). The reagents for the high resolution respirometry protocols were obtained from Sigma-Aldrich (St. Louis, MO, USA).

For the calculation of flux control ratios (FCR), absolute fluxes (per mg wet weight) were divided by the respective ETS-capacity [28].

### 2.3. Statistics

Two-way ANOVAs (genotype x age) were calculated to compare respiration or flux control ratios at different states. In case of significant interaction effects, Tukey’s post-hoc tests were calculated. SPSS was used to calculate statistical tests, Graphpad Prism was used for the generation of graphs.

## 3. Results

Our R6/2 mice displayed pronounced HD-related symptoms at 10–11 weeks of age, as reported previously [29,30,31]. These symptoms included motor dysfunction, as assessed by Rotarod, horizontal ladder task and limb clasping tests, reduced brain and body weight gain, and aggregated mHtt in the striatum.

For this study, pre-symptomatic (4 weeks) and symptomatic (10–11 weeks) R6/2 mice as well as wild-type mice of the respective same age (four animals per group) were sacrificed. The striatum, hippocampus, and motorcortex were dissected, transferred into preservation buffer (MiR06), and immediately used for high resolution respirometry after permeabilization, as previously reported [26,32]. The addition of substrates for mitochondrial Complex I in the absence of adenosine diphosphate (ADP, P) allowed us to monitor the nicotinamide adenine dinucleotide (NADH, N)-linked LEAK state N(L). The addition of ADP yielded the NADH-linked oxidative phosphorylation (OXPHOS) state N(P), and further supplementation with succinate (S) resulted in the NS(P) state. The subsequent uncoupling with FCCP induced the electron transfer system (E) capacity NS(E), followed by S(E) after the injection of Complex I inhibitor rotenone. Complex IV activity was assessed for each preparation by the addition of *N,N,N′,N′*-tetramethyl-*p*-phenylenediamine (TMPD) and ascorbic acid.

We observed a higher absolute respiration (per fresh weight) in 10–11-week-old mice as compared to 4-week-old mice in almost all respirational states in the striatum (Figure 1A), only for N(P) and NS(P) in the hippocampus (Figure 1B) and no age-related effects in the motor cortex (Figure 1C). There was no difference in any respirational state or analyzed brain region with respect to genotype (wild-type versus R6/2). See Appendix A for statistical results.

We next normalized the respirational values to the respective preparations’ electron transfer system maximum capacities NS(E), yielding flux control ratios (FCR) (Figure 2). N(L) FCRs were higher in 4-weeks-old than in 10–11-week-old animals in all analyzed brain regions. In contrast, N(P) and NS(P) were higher in the older animals in the striatum but not in the other brain regions. S(E) on the other hand was reduced in the older animals in both the striatum and motor cortex, with no detectable effects in the hippocampus. In the motor cortex, we, in addition, observed an interaction effect between age and genotype, indicating significantly more reduction of S(E) with age in wild-type as compared with R6/2 mice. See Appendix A for numerical results statistical results and Appendix A for detailed statistical assessment.

## 4. Discussion

In this study, we report significant brain region-dependent shifts in mitochondrial respirational patterns from juvenile to adult age, confirming our principal hypothesis. These shifts, however, for the largest part, were not different in R6/2 mice, indicating that in these HD model mouse, tissues’ compensatory mechanisms may be able to maintain overall respiration.

Absolute respiration (oxygen flux per fresh tissue weight) was significantly increased from 4 weeks to 10–11 weeks of age for all investigated respirational states (except for the LEAK state N(L)) in the striatum. This effect was limited to mitochondrial Complex I and Complex I & II-linked respiration in the hippocampus and entirely absent in the cortex.

The increased electron transfer system capacity NS(E) and Complex IV activity in adult striatum indicate that this effect is linked to elevated general mitochondrial respiratory capacity, a marker for mitochondrial density [26]. Overall respiratory capacity seems not to increase from juvenile to adult state in both the hippocampus and cortex.

FCRs give information on respiratory control independent of maximal electron transfer system capacity and, thus, are theoretically independent of mitochondrial content. We observed a lower relative LEAK N(L) state in all investigated brain regions in juvenile mice, indicating lower proton-LEAK and, thus, potentially tighter coupling in adult brain tissues. Relative succinate-driven S(E) contributions to maximum respirational capacities of striatum and cortex were reduced in adulthood, suggesting a higher reliance of these brain regions on Complex II at younger age.

Our work confirms previous reports of no [25] or marginal [24] impairment of striatal respiration in symptomatic R6/2 mice. We expand these results by demonstrating that also the hippocampus and motor cortex respiration are not impaired in these mice (within the limitations of the applied respirometry methods as discussed below). Furthermore, we show that transitions of respiration patterns from juvenile to adult mice are not affected by human Htt overexpression and the onset of HD-related symptoms, with the possible exception of succinate-linked respiration S(E) FCR in the cortex (Figure 2A). While the cortical S(E) FCR was reduced similarly to the striatum in adulthood, it was already at comparably low levels in R6/2 pre-symptomatic mice, suggesting early reductions in succinate-linked respiration in R6/2 mice. A summary of the brain-region dependent differences in FCRs as a function of age and genotype (WT versus R6/2) is presented in Figure 3.

Taken together, our results demonstrate clear brain-region-specific changes in mitochondrial respirational capacities as assessed by high resolution respirometry from juvenile to adult age that were only slightly affected in R6/2 mice in cortical succinate-driven respiration. It is important to consider that our results—and the greatest part of studies investigating mitochondrial respiration in brain regions—reflect the maximum capacities of the mitochondrial preparations, meaning that oxygen and substrate concentrations were maintained at saturating levels. Therefore, we cannot exclude oxygen or substrate limitations that may have had an impact on respiration in vivo but are masked in our ex vivo approach. In particular, our recent results on the efficiency of promoting sphingosine-1-phosphate (S1P) in the same R6/2 model [30] highlight the possibility of problems of oxygen availabilities in the R6/2 model and in HD. S1P has been shown to modulate hypoxia signaling by activating hypoxia inducible factor 1 (HIF-1) [32,33,34], thus, possibly exerting beneficial effects in HD disease progression via activating protective hypoxia-related pathways. Recently, the sphingolipid system has been increasingly linked to aging and may be importantly involved in neurodegenerative disease [35,36]. Future research elucidating the interplay of the sphingosine system with oxygen levels and oxygen extraction by mitochondria in aging model organisms of neurodegeneration and patients, therefore, has the potential to significantly advance our understanding of selective regional brain vulnerabilities in neurodegenerative diseases.

Furthermore, the role of substrates for oxidative phosphorylation and the interplay with other cellular pathways of energy production (e.g., glycolysis and the creatine phosphate shuttle) in neurodegenerative diseases are insufficiently explored yet. Importantly, respirational deficits in aged mice have previously been linked to specific substrates [37] and in HD neurons may specifically display mitochondrial deficits in energy-demanding conditions [38].

Our results indicate brain region-specific dynamic changes of respirational capacities with advancing age even in the absence of pathology that warrant further investigation. Observations of possibly different patterns of respirational changes in our previous studies [26,27,39] in various brain regions of older (around 3–6 months old) male control mice warrant more research on the development of mitochondrial respiration control in dependence of older age and sex. Such experiments may have pronounced implications on normal aging-related vulnerabilities of specific brain circuits and potential pathogenesis of various neurological diseases, also with regard to gender-specific differences.

In conclusion, we demonstrate the feasibility of detecting age-related changes in the respirational capacities of different brain regions using high resolution respirometry. In line with previous reports, the deficits of respiration of R6/2 mice were small, including the novel results on brain region-specific, age-dependent changes provided here. We argue that respirometric analyses are not well suited to uncover in situ oxygen conditions and are limited in the detection of potential substrate limitations but, in particular, these aspects of mitochondria may be important to understand brain-regional vulnerability in HD. They did, however, allow us to observe respirational shifts that are particularly pronounced in the striatum, but not in the hippocampus. The subtle difference of developmental respirational alterations in R6/2 mice in succinate driven respiration in the motor-cortex may be an important factor in HD pathogenesis that warrants further investigation.

## Figures and Tables

**Figure 1 ijms-21-05412-f001:**
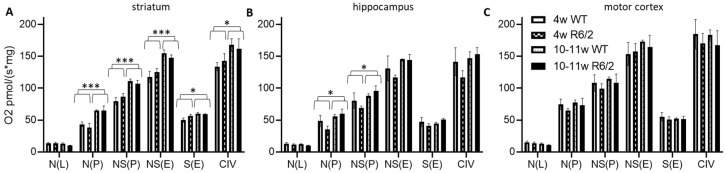
High resolution respiration data per mg wet weight. Striatum (**A**), hippocampus (**B**), and motor cortex (**C**) of wild-type (WT) and R6/2 mice before (4 weeks old, 4w) and after (10–11 weeks old, 10–11w) onset of motor symptoms. N = 4 animals per group. Means and SD are depicted. Two-way ANOVAs were calculated to test for statistical differences. * *p* < 0.05, *** *p* < 0.001.

**Figure 2 ijms-21-05412-f002:**
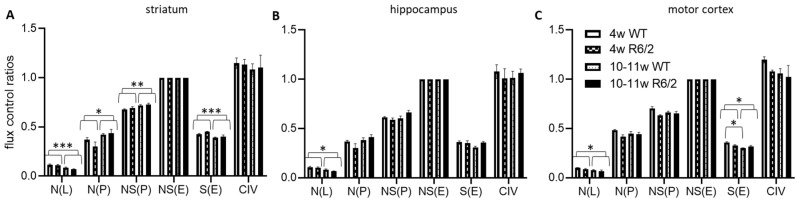
High resolution respiration data normalized to electron transfer system capacity. Striatum (**A**), hippocampus (**B**), and motor cortex (**C**) of wild-type (WT) and R6/2 mice before (4 weeks old, 4w) and after (10–11 weeks old, 10–11w) onset of motor symptoms. *n* = 4 animals per group. Means and SD are depicted. Two-way ANOVAs and Tukey’s post-hoc tests (in case of significant interaction effects) were calculated to test for statistical differences. * *p* < 0.05, ** *p* < 0.01, *** *p* < 0.001.

**Figure 3 ijms-21-05412-f003:**
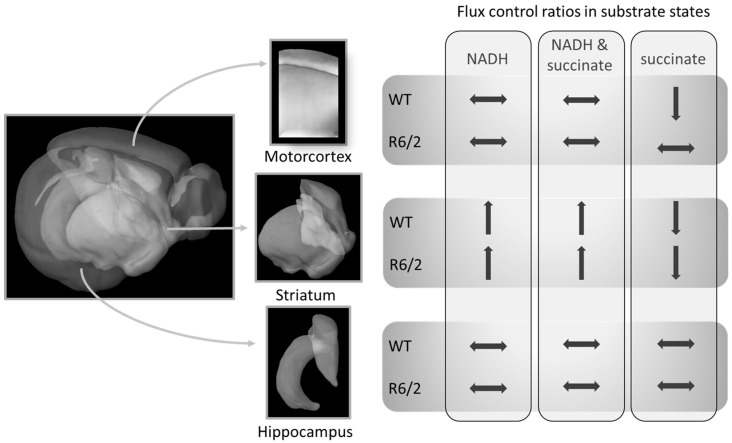
Relative mitochondrial respiration (flux control ratios) changes from 4- to 10–11-week-old mice differently in specific brain regions. These shifts in R6/2 were only altered in succinate-driven respiration as compared to wild-type (WT) mice. Brain Explorer 2 was used for the generation of brain region images.

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
