# Peer review of "Mitochondrial Respiration Changes in R6/2 Huntington’s Disease Model Mice during Aging in a Brain Region Specific Manner"

_ijms, 2020, doi:10.3390/ijms21155412_

Round 1

Reviewer 1 Report

The authors use ex-vivo high resolution respirometry to quantify oxidative phosphorylation across three brain regions (hippocampus, cortex and striatum) at 4 weeks and 10-11 weeks of age in the R6/2 mouse model of Huntington’s Disease and wild-type controls. Mitochondria dysfunction plays a key role in neurodegeneration and also normal ageing. The R6/2 model of Huntington’s is reasonably well established, although as the authors point out data on mitochondrial respiration for this model is conflicting. In this sense the study presented is important. The study is simple and well designed, well conducted, carefully presented and cautiously discussed – with disclosure of potential limitations. The changes in OXPHOS reported by the authors are relatively small / subtle but they are nevertheless interesting and should be published.

I only have very minor comments:

  • I would have liked to see OSPHOS profiling in older mice – (>12-14 months). I appreciate that this is not feasible in the R6/2 model. Perhaps the authors have additional data from older WT mice they could incorporate to their discussion / supplementary?
  • Table 1 might be more suitable as supplementary table. I would also like to see a table with the actual values/SD displayed in the graphs on figure 1. This is interesting information and the changes reported are subtle and trying to work out the actual O2 pml/(s*mg) from the graphs is not particularly accurate. This could be a supplementary table.

Author Response

Reviewer 1:

The authors use ex-vivo high resolution respirometry to quantify oxidative phosphorylation across three brain regions (hippocampus, cortex and striatum) at 4 weeks and 10-11 weeks of age in the R6/2 mouse model of Huntington’s Disease and wild-type controls. Mitochondria dysfunction plays a key role in neurodegeneration and also normal ageing. The R6/2 model of Huntington’s is reasonably well established, although as the authors point out data on mitochondrial respiration for this model is conflicting. In this sense the study presented is important. The study is simple and well designed, well conducted, carefully presented and cautiously discussed – with disclosure of potential limitations. The changes in OXPHOS reported by the authors are relatively small / subtle but they are nevertheless interesting and should be published.

We thank the reviewer for appreciating design, experimental approaches, presentation and discussion of our study and in particular for highlighting the discussed limitations, which we think are important to move the field further.

I only have very minor comments:

  • I would have liked to see OSPHOS profiling in older mice – (>12-14 months). I appreciate that this is not feasible in the R6/2 model. Perhaps the authors have additional data from older WT mice they could incorporate to their discussion / supplementary?

We agree that these are interesting perspectives to add to the study. As pointed out by the reviewer, the assessment of OXPHOS in older R6/2 mice is not feasible. The R6/2 mice we use (carrying 160 CAG) die already at 12 – 14 weeks of age. At 11 weeks of age their motoric function is already massively impaired. Moreover, data from old healthy animals probably have little relevance for disease pathogenesis considering that neuronal and lipid metabolism deficits induce secondary mitochondrial dysfunctions probably related to general impairment of brain homeostasis.
However, to address this point, we now added a discussion of findings of our prior studies using older – but male – mice (lines 227-234, highlighted). However, a clear evaluation of age- (and sex-) related differences in mitochondrial respiration control requires well-designed, tightly controlled independent research. This is based on our observation that many factors have the capacity to pronouncedly affect mitochondrial respiration also apart from age- and sex-differences in the same species; e.g. slight optimizations in substrate doses (e.g. succinate), the equipment used for tissue permeabilization and other conditions for example depending on localization, circadian rhythms and animal husbandry, increasing inter-experimental noise, despite very robust intra-experimental reliability.

Table 1 might be more suitable as supplementary table. I would also like to see a table with the actual values/SD displayed in the graphs on figure 1. This is interesting information and the changes reported are subtle and trying to work out the actual O2 pml/(s*mg) from the graphs is not particularly accurate. This could be a supplementary table.

Thank you for these suggestions: we agree and moved the former table 1 into the supplements now as supplementary table 2. We also included the proposed table summarizing the data of our respiration in numerical form. It is presented as supplementary table 1 now (references in main text lines 162-163, highlighted).

Reviewer 2 Report

This is an interesting study. The authors analyzed the brain region-specific alterations of oxidative phosphorylation, and determined age-related changes in respirational capacities of different brain regions in R6/2 HD mice. I have only minor concerns that should be addressed by the authors:

  1. The title is confusing. What is “developmental respirational shifts”? Please consider making proper revision, so readers can easily understand what your work is about.
  2. In Introduction part line 47, the authors claimed “Strikingly, in most neurodegenerative diseases, specific brain regions are selectively affected”. Please add sufficient references to support your argument.
  3. Are there any sex differences for brain-specific mitochondrial dysfunction in R6/2 mice?
  4. My concern is 10-11 weeks of R6/2 mice and WT mice are still young mice, even though there are pathological changes in HD R6/2 mice. Did the authors use old mice, for example 12 or even 18-month mice to determine the mitochondrial respiration alteration?

Author Response

Reviewer 2 :

This is an interesting study. The authors analyzed the brain region-specific alterations of oxidative phosphorylation, and determined age-related changes in respirational capacities of different brain regions in R6/2 HD mice. I have only minor concerns that should be addressed by the authors:

We thank reviewer 2 to point out the interest of our study for the field!

1. The title is confusing. What is “developmental respirational shifts”? Please consider making proper revision, so readers can easily understand what your work is about.

Thank you for making us aware of this ambiguity in the title. We think that the new title “Mitochondrial respiration changes in R6/2 Huntington`s Disease model mice during aging in a brain region specific manner” reflects the contents in a clearer way.

2. In Introduction part line 47, the authors claimed “Strikingly, in most neurodegenerative diseases, specific brain regions are selectively affected”. Please add sufficient references to support your argument.

We agree with the reviewer that references were missing for this part. A new reference to the review on “Selective vulnerability in neurodegenerative diseases” by Fu, Hardy and Duff (2018, Nat Neurosci) has now been inserted (line 50, highlighted). In this work also the mentioned brain regional vulnerabilities in different neurodegenerative diseases are excellently presented.

3. Are there any sex differences for brain-specific mitochondrial dysfunction in R6/2 mice?

Thank you for this important question. Due to fertility problems most male R6/2 are used for the breeding, therefore we are very limited in terms of experimental male R6/2. In our experience there are no big differences between male and female mice in terms of molecular and biochemical pathways (we tested this at late stages of the disease). This of course does not exclude that there are sex-dependent differences in mitochondrial respiration. We think that it will be especially important to now conceive experiments specifically designed to evaluate sex- and age-related mitochondrial respiration changes in different brain regions in healthy animals. We now address this point in the discussion (lines 227-234, highlighted).

4. My concern is 10-11 weeks of R6/2 mice and WT mice are still young mice, even though there are pathological changes in HD R6/2 mice. Did the authors use old mice, for example 12 or even 18-month mice to determine the mitochondrial respiration alteration?

Thank you for raising this point – yes, the 10-11 weeks reflect young adult mice. This age was chosen due to ethical concerns (stronger symptom development, in particular major motor impairment already by 11 weeks of age) and because the R6/2 mice we use (carrying 160 CAG) die already at 12 – 14 weeks of age. Furthermore, potential confounders at later age complicate the interpretation of the data: severe neuronal and lipid metabolism deficits induce secondary mitochondrial dysfunction that is probably more related to general impairment of brain homeostasis than to the HD mutation and therefore may have little relevance for disease pathogenesis. We now mention the need to systematically investigate age-dependent changes in healthy mice in the discussion (lines 227-234, highlighted).